# Improving Model Compatibility of Generative Adversarial Networks by Boundary Calibration

## Abstract

Generative Adversarial Networks (GANs) is a powerful family of models that learn an underlying distribution to generate synthetic data. Many existing studies of GANs focus on improving the realness of the generated image data for visual applications, and few of them concern about improving the quality of the generated data for training other classifiers—a task known as the model compatibility problem. Literature also show that some GANs often prefer generating 'easier' synthetic data that are far from the boundaries of the classifiers, and refrain from generating near-boundary data, which are known to play an important roles in training the classifiers. To improve GAN in terms of model compatibility, we propose Boundary-Calibration GANs (BCGANs), which leverage the boundary information from a set of pre-trained classifiers using the original data. In particular, we introduce an auxiliary Boundary-Calibration loss (BC-loss) into the generator of GAN to match the statistics between the posterior distributions of original data and generated data with respect to the boundaries of the pre-trained classifiers. The BC-loss is provably unbiased and can be easily coupled with different GAN variants to improve their model compatibility. Experimental results demonstrate that BCGANs not only generate realistic images like original GANs but also achieves superior model compatibility than the original GANs.

## 1 Introduction

The success of machine learning relies on not only the advances of different models (e.g. deep learning) but also data with sufficient quality and quantity. Nowadays, companies spend tremendous efforts and expense collecting data to build their products. To better solve complicated real-world problems with public or third-party machine learning experts, many companies now needs release some data sets for competitions (e.g. Kaggle) or proof-of-concept purposes. However, considering the costs of collecting data, companies may not be willing to release the dataset if possible. As a result, a technique which can generate synthetic data with properties similar to the original data is in demand. To be specific, we are looking for generating a dataset with the property that machine learning models trained on the generated dataset can exhibit similar performance to ones trained on the original data. This property is called *model compatibility* (Park et al., 2018) or *machine learning efficacy* (Xu et al., 2019). The organizations can share the generated data with high model compatibility to the public and enjoy the solution derived from it without leaking the real dataset.

When it comes to data generation, generative adversarial networks (GANs, Goodfellow et al. 2014) is the most popular family of generative algorithms because of its impressive performance on generating realistic images (Karras et al., 2018). In GANs, the generator is trained via minimizing a neural network (discriminator) defined probability divergence (Goodfellow et al., 2014; Arjovsky et al., 2017; Nowozin et al., 2016). In addition to image generation, GANs are also widely used in other applications, such as style transfer (Isola et al., 2017; Zhu et al., 2017; Kim et al., 2017) and image processing (Pathak et al., 2016; Ledig et al., 2017; Chang et al., 2017), and generating different types of data, including time series (Luo et al., 2018; Chang et al., 2019), text (Yu et al., 2017; Press et al., 2017), point clouds (Li et al., 2018), voxels (Wu et al., 2016) and tabular data (Park et al., 2018; Xu et al., 2019).

Although GANs are versatile as mentioned above, most of their development focus on the metrics such as quality and diversity of the data Salimans et al. (2016); Heusel et al. (2017); Lucic et al. (2018). Generating high model compatibility data via GANs is still under explored. The pioneering work (Xu et al., 2019) first shows that data generated from conditional GANs (Mirza & Osindero, 2014) enjoys better model compatibility than VAEs (Kingma & Welling, 2013). So we wonder **can we improve the model compatibility if we consider information from the models trained on the original data?** For example, Wasserstein GAN (WGAN, Arjovsky et al. 2017) performs a mean-matching between the distribution of real data and generated data. However, only mean-matching is sometimes not enough to learn the whole distribution especially for those boundary cases. Apparently, if a GAN knows the boundary between different classes, it may be able to generate instances which are close to the boundary with correct labels. These boundary points will guide a classifier to learn the correct decision boundary.

In this work, we try to improve GANs with regards to model compatibility in classification problems. We use a set of pre-trained classifiers to obtain multiple decision boundaries. Then use an auxiliary loss function called Boundary-Calibration loss (BC-loss) to calibrate the generating distribution according to the decision boundaries of these pre-trained classifiers. The main contributions of this work are:

- In Section 2, we propose a way to evaluate model compatibility in classification problems. We consider a variety of machine learning algorithms and average the performance to obtain a comprehensive metric.

- In Section 4, we propose a loss function called Boundary-Calibration loss (BC-loss) which helps typical GANs to learn a distribution with better model compatibility. The loss considers the decision boundaries of pre-trained classifiers and minimizes the maximum mean discrepancy (MMD, Gretton et al. 2012) between the original dataset and the generated dataset. In addition, we show that optimizing the BC-loss would not change the optimal solution of the original GAN, but it reduces the feasible set to ensure the model compatibility.

- In Section 5, we demonstrate how BC-loss affects the boundary of the generated data with a two-dimensional toy dataset. We also show that the BC-loss improves model compatibility of the generated data with different types of classifiers and a variety of datasets. Finally, we inspect the feature selection results to examine how the interpretation of machine learning models may be effected.

Last, in Section 3, we discuss some works which are similar to our work and describe how does our work differ from them.

## 2 MODEL COMPATIBILITY IN CLASSIFICATION

In this work, we focus on generating data for fully-supervised classification learning. Given a dataset $D = \{(\boldsymbol{x}_i, y_i)\}_{i=1}^n$, where $\boldsymbol{x}_i \in \mathcal{X}$ represents features of an instance, $y_i = f(\boldsymbol{x}_i) \in \mathcal{Y}$ represents the corresponding label of $\boldsymbol{x}_i$ according $f : \mathcal{X} \rightarrow \mathcal{Y}$, and $(\boldsymbol{x}_i, y_i) \sim P_D$, a learning algorithm $A : (\mathcal{X}, \mathcal{Y})^n \rightarrow \mathcal{H}$ learns a hypothesis $h \in \mathcal{H}$ to approximate the mapping function, i.e. $A(D) = h \approx f$. Our goal is to obtain a generator $G$ which generates a synthetic dataset $D' = \{(\boldsymbol{x}'_j, y'_j)\}_{j=1}^m$ such that $A(D') = h' \approx h$. We call this property model compatibility as proposed in Park et al. (2018).

To measure the model compatibility of a generated dataset quantitatively, we consider the performance of a classifier trained on the generated dataset comparing to the one trained on the real dataset. We evaluate the accuracy on a separate test dataset to indicate the performance of a given classifier. In addition, we calculate relative accuracy by scaling the test accuracy of the classifier trained on the generated dataset by the accuracy of the classifier trained on the real dataset. The relative accuracy allows us to average the results from different machine learning algorithms more fairly. The final evaluation is :

$$\frac{1}{|\mathbb{A}|} \sum_{A \in \mathbb{A}} \frac{acc(h', D^{(t)})}{acc(h, D^{(t)})}, \tag{1}$$

where $\mathbb{A}$ is a set of learning algorithms , $D^{(t)} = \{(x_i^{(t)}, y_i^{(t)})\}_{i=1}^N$ is the test dataset, and $acc(h, D^{(t)})$ is the accuracy of hypothesis $h$ on test data $D^{(t)}$. We can determine $\mathbb{A}$ as a wide variety of learning algorithms to make the metric provide a more comprehensive measurement of model compatibility.

## 3 RELATED WORK

Research about generating data for classification can be divided into two categories: formulation and architecture. For formulation, Conditional GAN (CGAN, Mirza & Osindero 2014 ) is an intuitive way to generate instances with corresponding labels. We can learn the distribution of labels by counting and sample the instances from CGAN conditionally. Auxiliary Classifier GAN (ACGAN, Odena et al. 2017) is considered as a better way for conditional generation. It uses an auxiliary classifier to provide information about the boundary between each classes. However, ACGAN has been proved that the objective is biased so it tends to generate data with lower entropy for the auxiliary classifier (Shu et al., 2017). Thus, the lose of instances near the decision boundary may worsen the model compatibility. In this work, we use CGAN along with the proposed BC-loss to generate data with model compatibility.

On the other hand, the other line of research focuses on generating data with different network architecture or data processing procedure. Recent works that also consider model compatibility are Table GAN (Park et al., 2018) and Tabular GAN (Xu et al., 2019). Table GAN focuses on the privacy of generated data and thus their is a trade off between privacy and model compatibility. To achieve privacy preserving, they do not improve the model compatibility compared to the original GAN. On the other hand, Tabular GAN puts emphasis on increasing model compatibility of generated data. They propose a framework with a more complicated data processing procedure and use LSTM to better parameterize the target distribution. In contrast to these works, our work focus on the formulation of GANs and can be applied to most variants of GANs, including Table-GAN and Tabular GAN. Moreover, while these former works only focus on tabular data, our BC-loss is applicable to generate image datasets as well.

Some GAN variants are named similarly to our work but they pay attention to different problems. For example, the boundary described in boundary-seeking GAN (Hjelm et al., 2017) means the decision boundary of the discriminator rather than the decision boundary for the supervised labels. To the best of our knowledge, we are the first work trying to improve model compatibility by modifying the formulation of GANs.

## 4 BOUNDARY-CALIBRATION GAN

To achieve better model compatibility of GAN, we propose an auxiliary GAN loss which we call boundary-calibration loss (or BC-loss). We assume that we have a set of pre-trained classifiers which are well-trained on the original dataset. The BC-loss helps GANs to calibrate the distribution with respect to the distribution of decision values predicted by pre-trained classifiers. The calibration leads to more accurate data generation near the decision boundary and thus enabling a machine learning algorithm to learn a similar hypothesis to one that learns from the original dataset. To ease the complexity of learning to generate $(\boldsymbol{x}, y)$ jointly, we infer $P(y)$ by counting the proportion of each class in the original dataset and train a conditional generator $G$ such that $G(\boldsymbol{z}, y) \sim P_{\mathcal{X}|y}$, where $P_{\mathcal{X}|y}$ is the conditional data distribution and $\boldsymbol{z} \sim P_{\mathcal{Z}}$ is the initial randomness such as Gaussian distribution. Therefore we can generate $(\boldsymbol{x}, y)$ by sampling $y \sim P(y)$ and $G(\boldsymbol{z}, y)$.

### 4.1 BOUNDARY CALIBRATION

Given a pre-trained classifier $C$, we hope the generated dataset will adopt the same statistics as the original dataset while considering the decision boundary of $C$. To include the information about the boundary, we calculate posterior $P_C(y \mid \boldsymbol{x}_i)$ from the decision values predicted by the classifier. In practice, we can apply a softmax function to the outputs of a classifier to obtain the posterior. The posterior provides information of an instance from the classifier's aspect. Therefore, given the real dataset $X = \{\boldsymbol{x}_1, \boldsymbol{x}_2, ..., \boldsymbol{x}_n\}$, we can obtain a set of posterior vector $C(X) = (P_C(y \mid \boldsymbol{x}_1), P_C(y \mid \boldsymbol{x}_2), ..., P_C(y \mid \boldsymbol{x}_n))$. To generate data $X'$ with the same distribution of posteriors to the boundary,

we match the statistics of $C(X)$ and $C(X')$ by optimizing a distance $M$:

$$\mathcal{L}_{BC}(X, X', C) = M(C(X), C(X')) \tag{2}$$

Here $M$ can be any distance metric which measures the distance between two sets of samples. In statistics, distinguishing whether two sets of samples are from the same distribution is called *Two-Sample Test*. A classical solution to two-sample test is kernel maximum mean discrepancy (MMD, Gretton et al. 2012). The idea is to compare the statistics between the two sets of samples. If the statistics are similar then these two sets might be sampled from the same distribution. Given two sets of samples $X = \{\boldsymbol{x}_i\}_{i=1}^n$ and $Y = \{\boldsymbol{y}_j\}_{j=1}^n$, an unbiased estimator of MMD with kernel $k$ is defined as:

$$\hat{M}_k(X, Y) = \frac{1}{\binom{n}{2}} \sum_{i \neq i'} k(\boldsymbol{x}_i, \boldsymbol{x}_{i'}) - \frac{2}{\binom{n}{2}} \sum_{i \neq j} k(\boldsymbol{x}_i, \boldsymbol{y}_j) + \frac{1}{\binom{n}{2}} \sum_{j \neq j'} k(\boldsymbol{y}_j, \boldsymbol{y}_{j'}) \tag{3}$$

In practice, we use Gaussian kernel $k(\boldsymbol{x}, \boldsymbol{x}') = \exp(\|\boldsymbol{x} - \boldsymbol{x}'\|^2)$ in MMD since Gaussian kernel is a characteristic kernel which ensures that the distance is zero if and only if the two distributions are the same (Gretton et al., 2012).

The BC-loss can be applied in generator of any GAN variants to improve the model compatibility. In addition, to better fit the real unknown boundary, we can use multiple classifiers to calibrate the distributions from different aspects. As a result, for a loss function of generator $\mathcal{L}_G$, we can modify the loss to be:

$$\hat{\mathcal{L}}_G = \mathcal{L}_G + \frac{\lambda}{|\mathbb{C}|} \sum_{C \in \mathbb{C}} \hat{M}_k(C(X), C(G(\mathbb{Z}, \mathcal{Y}))) \tag{4}$$

where $\mathbb{Z}$ is a set of noises, $\mathbb{C}$ is a set of pre-trained classifiers and $\lambda$ is a hyper-parameter to control the weight of BC-loss.

## 4.2 ANALYSIS OF OPTIMAL SOLUTION

Next we prove that adding our proposed BC-loss would not change the optimal solution of the original objective. Here we assume the loss of the generator $\mathcal{L}_G$ achieves its optimal value in the its GAN objectives $\mathcal{L}_G$ if and only if the distribution of $G(\boldsymbol{z}, y)$ recovers $P_{\mathcal{X}|y}$ for all $y \in \mathcal{Y}$, which holds for the vanilla GAN (Goodfellow et al., 2014) and most of other GAN variants.

**Theorem 1** (Gretton et al. 2012). *Given a kernel $k$, if $k$ is a characteristic kernel, then $M_k(P, Q) = 0 \iff P = Q$.*

**Theorem 2** (Equivalence of optimal solution). *$G$ is an optimal solution of $\mathcal{L}_G \iff G$ is an optimal solution of $\hat{\mathcal{L}}_G$*

*Proof.* ($\Rightarrow$) According to the assumption, $G$ is an optimal solution of $\mathcal{L}_G$ implies $G(\mathcal{Z}, y)$ recovers $P_{\mathcal{X}|y}$ for all $y \in \mathcal{Y}$. Therefore, $P_{C(X)} = P_{C(G(\mathbb{Z}, \mathcal{Y}))}$ and $M_k(P_{C(X)}, P_{C(G(\mathbb{Z}, \mathcal{Y}))}) = 0$ by Theorem 1. Now $\hat{\mathcal{L}}_G = \mathcal{L}_G + 0 = \mathcal{L}_G$ and $G$ is an optimal solution of $\mathcal{L}_G$, so $G$ is also an optimal solution of $\hat{\mathcal{L}}_G$.

($\Leftarrow$) Since $\mathcal{L}_{BC} \geq 0$, we have $\hat{\mathcal{L}}_G = \mathcal{L}_G + \mathcal{L}_{BC} \geq \mathcal{L}_G$. From above, we know $\hat{\mathcal{L}}_G(G) = 0$ if $G = P_{\mathcal{X}}$. Thus, for an optimal solution $G^*$, $0 \geq \hat{\mathcal{L}}_G(G^*) \geq \mathcal{L}_G(G^*) \geq 0$, which implies $\hat{\mathcal{L}}_G(G^*) = \mathcal{L}_G(G^*) = 0$. Therefore, $G^*$ is also an optimal solution of $\mathcal{L}_G$. $\square$

The proof shows that the proposed BC-loss does not change the optimal solution of the original optimization problem. However, we can consider BC-loss as a Lagrangian constraint which restricts the solution to a subspace where the generator owns higher model compatibility .

## 4.3 COMPARISON TO MMD GAN

MMD GAN (Li et al., 2017) is a variant of GAN where the generator tries to minimize the MMD between generated data and original data and the discriminator learns a kernel which maximizes the MMD. Though the formulation of MMD GAN and BC-loss are similar, they still do not conflict because MMD GAN do not known the information about the classifier and the objective of

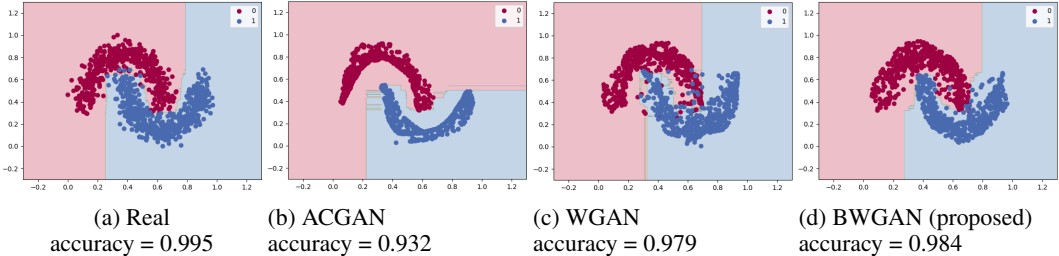

|  (a) Real | (b) ACGAN | (c) WGAN | (d) BWGAN (proposed) |



(a) Real
accuracy = 0.995

(b) ACGAN
accuracy = 0.932

(c) WGAN
accuracy = 0.979

(d) BWGAN (proposed)
accuracy = 0.984



Figure 1: A toy dataset generated by different GAN methods. Figure (a) is the original training data and the others are data generated by ACGAN, WGAN and our BWGAN respectively. The background color indicates the decision boundary of a random forest trained on corresponding data. The captions show the test accuracy of the random forest.

MMD GAN would not lead the discriminator to a classifier. Therefore, BC-loss may still improve MMD GAN by guiding the generator to not generate points across the boundary. To understand the improvement in MMD GAN from BC-loss, we use MMD GAN as one of the baselines in our experiments.

## 5 EXPERIMENTS

In this section, we use a toy dataset to illustrate how the proposed method improves the model compatibility. To be more realistic, we provide more comprehensive results for four different real-world dataset from UCI dataset repository (Dua & Graff, 2017): Adult, Connect-4, Covertype and Sensorless. We then show our method is also applicable in image dataset: MNIST and Cifar10 without losing the image quality. In addition, we investigate the results of feature selections on the generated dataset to see whether the generated data can preserve the interpretation of machine learning models.

### 5.1 EXPERIMENTAL SETTINGS

#### EVALUATION

In this work, we focus on model compatibility of generated datasets. We use a wide variety of machine learning algorithms including linear SVM, decision tree (DT), random forest (RF), and multi-layer perception (MLP) to evaluate the model compatibility. As described in Section 2, we evaluate the relative accuracy for each type of machine learning model, where the relative accuracy is calculated by dividing the accuracy of classifier trained on generated data to the accuracy of classifier trained on original data.

#### COMPARED METHODS

We take Wasserstein GAN (WGAN) and MMD GAN as our baselines to evaluate the effectiveness of the proposed boundary-calibration technique. We denotes their counterparts with BC-loss as BWGAN and BMMDGAN respectively. All of the methods use gradient penalty to enforce the Lipschitz constraint on the discriminator (Gulrajani et al., 2017; Li et al., 2017). To achieve conditional data generation as described in Section 4, we add an embedding layer to learn the embedding vector for each class. The embedding vector is concatenated as additional input features for both generators and discriminators on UCI datasets. For image datasets, the embedding vectors are used as described in Mirza & Osindero (2014).

### 5.2 2D TOY DATASET

We use a 2D toy dataset with two classes to illustrate the results generated by different GAN methods in Figure 1. Figure 2a shows the distribution of the original training data. We use these generated data to train a random forest and depict the decision boundary by different background color. From Figure 2b, we can see that although ACGAN can make use of the auxiliary classifier during training,

Table 1: Summary result of model compatibility evaluate on UCI datasets. The numbers are relative accuracy.

|         | adult | connect4 | covertype | sensorless | average |
|---------|-------|----------|-----------|------------|---------|
| ACGAN   | 97.78 | 83.71    | 51.98     | 77.47      | 77.74   |
| WGAN    | 96.60 | 87.59    | 79.56     | 84.63      | 87.10   |
| BWGAN   | **98.79** | **88.95** | **83.16** | **93.34** | **91.06** |
| MMDGAN  | 95.67 | 86.29    | 77.14     | 86.28      | 86.35   |
| BMMDGAN | **97.23** | **87.38** | **79.82** | **88.14** | **88.14** |

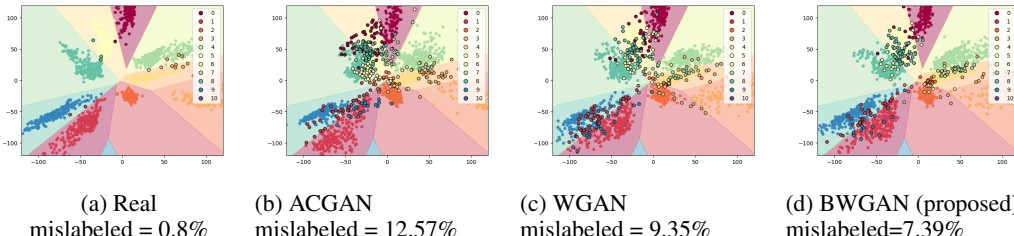

| (a) Real | (b) ACGAN | (c) WGAN | (d) BWGAN (proposed) |
|----------|-----------|----------|----------------------|
| mislabeled = 0.8% | mislabeled = 12.57% | mislabeled = 9.35% | mislabeled=7.39% |

Figure 2: 2D visualization of real and generated Sensorless dataset. The mislabeled points are emphasized with border lines. The background color indicates the spaces of each class according to the projection classifier.

it learns a biased distribution that push the generated data away from the boundary. The large margin between the two clusters brings more uncertainty to the decision boundary and thus leads to worse test accuracy. In Figure 2c, WGAN approximates the original distribution well in the center part of the two cluster, but do not get a clear boundary between the two classes. It generates some ambiguous points near the boundary that would confuse the classifier. Finally, our BWGAN generates points near the boundary more precisely, as shown in Figure 2c.

## 5.3  UCI DATASET

We evaluate our proposed BC-GAN on four datasets from UCI repository. The attributes of the datasets can be found in Appendix A. Discrete features are processed to one-hot encoding and continuous features are scaled to $[0, 1]$. For each dataset, we train six multi-layer perceptrons with a random split of half of training data as pre-trained classifiers. In these experiments, generators and discriminators are consist of 3 fully-connected hidden layer with 128 units. A logistic function is applied to the output layer of generators to generate features within 0 and 1. The weight of BC-loss is set to be $\lambda = 100$ for all datasets.

Table 1 summarize the comparison between different methods. We calculate the relative accuracy of different machine learning models mentioned in Section 5.1 and average the relative accuracy to indicate the model compatibility of generated data for each dataset. The table shows that the proposed BC-loss improves the accuracy of classifiers generally compared to original WGAN and MMD GAN. Moreover, ACGAN performs worst on three out of four datasets and exhibit a significant deficiency though it is proved to have the state-of-the-art generation quality. This again prove that the biased objective of ACGAN worsen the model compatibility seriously.The breakdown results and real accuracy are provided in Appendix B.

To further investigate the advantage of boundary-calibration, we visualize the generated results of Sensorless in Figure 2. We train a fully-connected neural network with a 2-units hidden layer before the output layer to project the generated samples to a 2-dimensional embedding space. The projection classifier is well-trained and achieves over 99% testing accuracy so we can use it to determine whether a sample is generated with incorrect label. The figure shows that there are less mislabeled data generated by BWGAN, especially at the center and the bottom-left region. The fact indicates that boundary-calibration helps GANs generate labeled data more accurately, which may lead to the improvement of classification accuracy.

Table 2: Precision at K of feature importance ranking compared to the feature importance ranking obtained from the original dataset

| dataset | metric | REAL | ACGAN | WGAN | BWGAN | MMDGAN | BMMDGAN |
|---------|--------|------|-------|------|-------|--------|---------|
| adult | P@10 | 0.80 | 0.40 | 0.80 | 0.80 | 0.80 | 0.70 |
| | P@20 | 0.95 | 0.45 | 0.75 | 0.80 | 0.85 | 0.75 |
| | P@30 | 1.00 | 0.63 | 0.90 | 0.90 | 0.87 | 0.87 |
| connect4 | P@10 | 0.90 | 0.30 | 1.00 | 1.00 | 0.90 | 0.80 |
| | P@20 | 0.90 | 0.35 | 0.90 | 0.85 | 0.95 | 0.85 |
| | P@30 | 0.93 | 0.33 | 0.83 | 0.90 | 0.77 | 0.83 |
| covertype | P@10 | 0.80 | 0.30 | 0.60 | 0.80 | 0.30 | 0.60 |
| | P@20 | 1.00 | 0.50 | 0.75 | 0.85 | 0.90 | 0.85 |
| | P@30 | 1.00 | 0.70 | 0.93 | 0.87 | 0.87 | 0.87 |
| sensorless | P@10 | 0.80 | 0.60 | 0.80 | 0.90 | 0.70 | 0.70 |
| | P@20 | 0.95 | 0.50 | 0.75 | 0.85 | 0.80 | 0.85 |
| | P@30 | 1.00 | 0.80 | 0.90 | 0.90 | 0.87 | 0.83 |

Table 3: F1 score of feature selection by $\ell_1$-regularized linear SVM

| dataset | metric | REAL | ACGAN | WGAN | BWGAN | MMDGAN | BMMDGAN |
|---------|--------|------|-------|------|-------|--------|---------|
| adult | f1 (C=0.01) | 0.975 | 0.571 | 0.697 | 0.787 | 0.795 | 0.725 |
| | f1 (C=0.001) | 0.968 | 0.500 | 0.737 | 0.789 | 0.700 | 0.789 |
| connect4 | f1 (C=0.01) | 0.905 | 0.860 | 0.889 | 0.866 | 0.874 | 0.831 |
| | f1 (C=0.001) | 0.933 | 0.718 | 0.796 | 0.739 | 0.750 | 0.752 |
| covertype | f1 (C=0.01) | 0.989 | 0.923 | 0.911 | 0.935 | 0.730 | 0.773 |
| | f1 (C=0.001) | 1.000 | 0.825 | 0.912 | 0.825 | 0.800 | 0.815 |
| sensorless | f1 (C=0.01) | 0.982 | 0.848 | 0.900 | 0.918 | 0.813 | 0.844 |
| | f1 (C=0.001) | 0.815 | 0.778 | 0.769 | 0.733 | 0.812 | 0.710 |

## 5.4 INTERPRETABILITY

In addition to accuracy, it is also important that the model trained on generated data should give us the same interpretation of a model trained on the original data. We investigate the interpretability by two common feature selection techniques. First, we train two random forests on the generated and original dataset respectively. Each random forest can provide the importances of the features. We evaluate the consistency of interpretation by calculating precision at Kth, which means how many features ranked top-k in random forest trained on original data are in the top-k importance feature of the random forest trained on generated data. The results are shown in Table 2. We provide the results of training a classifier on the same original data with a different random seed as *REAL* for comparison. The effect of BC-loss is not significant in this aspect. However, the scores of ACGAN drop seriously, which means training a classifier on data generated by ACGAN is somehow dangerous because the meaning of model may be totally different.

Another way to select feature is training a linear model with $\ell_1$ regularization. In Table 3 we use linear SVM with $\ell_1$ regularization to select features. Then we calculate the F1 score of features selected by classifiers trained on generated data to known how similar between the two sets of features selected by classifiers trained on original and generated dataset. The results again shows that using boundary-calibration does not has significant effect to feature selection and ACGAN is not proper to generated data for training.

## 5.5 IMAGE DATASET

We further use MNIST and CIFAR-10 dataset to investigate the effectiveness of boundary-calibration on image datasets. For MNIST, we train six 4-layer convolution neural networks (CNN) with random sampling half of training data as pre-trained classifiers, and use the same classifier set in Section 5.1 to evaluate model compatibility. For CIFAR-10, we use ResNet56v2 (He et al., 2016) to obtain three pre-trained classifier and evaluate on CNN and ResNet56v2. In both task, we use DCGAN (Radford et al., 2016) as network structure in all GANs. The weight of BC-loss is set to be $\lambda = 1$ for these two datasets.

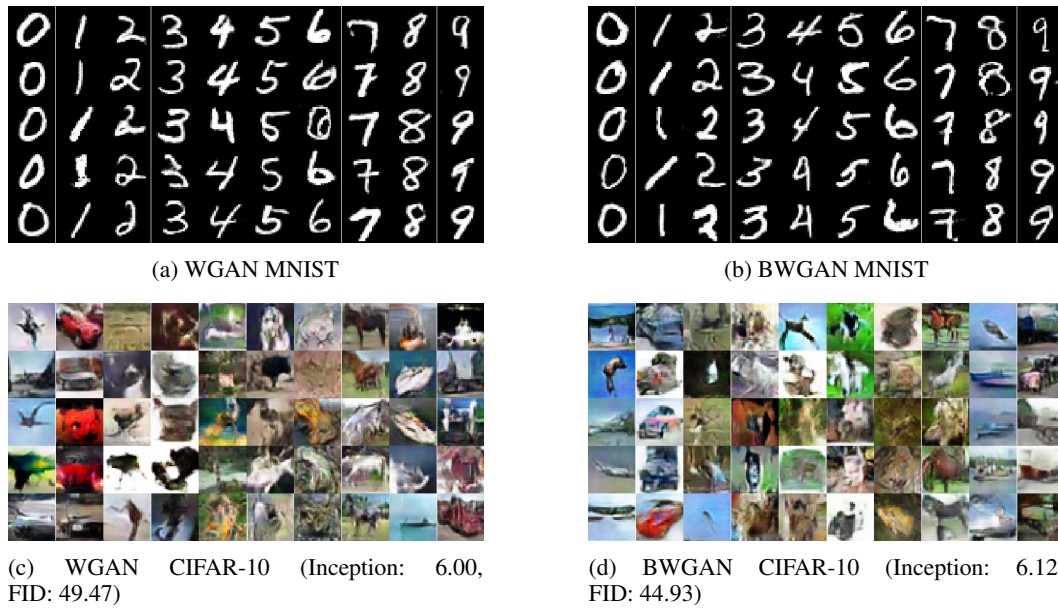

(a) WGAN MNIST

(b) BWGAN MNIST

(c) WGAN CIFAR-10 (Inception: 6.00, FID: 49.47)

(d) BWGAN CIFAR-10 (Inception: 6.12, FID: 44.93)

Figure 3: Gnerated samples from WGAN and our BWGAN. The images in the same column are in the same category.

Table 4 and Table 5 show the relative accuracy of classifiers trained on generated data. The proposed BWGAN still outperforms WGAN with better accuracy in general. The results generated by WGAN and BWGAN are pictured in Figure 3. The Inception score and Frechet Inception Distance (FID) for CIFAR-10 are also provided in the caption of Figure 3. Though the difference of quality between the images generated from WGAN and BWGAN is not significant in visual, the quantitative scores for quality of generated samples of CIFAR-10 are slightly improved. The results indicate that even though our method seems not improve the image quality, it is still able to improve the model compatibility without losing image quality.

Table 4: Breakdown results on MNIST dataset.

|               | REAL  | WGAN        | BWGAN        |
|---------------|-------|-------------|--------------|
| DT (d=10)     | 86.6  | 54.5 (63.0) | 48.3 (55.8)) |
| DT (d=20)     | 88.0  | 34.5 (39.2) | 48.2 (54.8)  |
| Linear SVM    | 88.0  | 34.5 (39.2) | 48.2 (54.8)  |
| MLP (100)     | 97.6  | 96.2 (98.5) | 96.8 (99.2)  |
| MLP (200x2)   | 97.9  | 96.7 (98.8) | 96.1 (98.2)  |
| RF (n=10, d=10) | 92.5 | 75.5 (81.7) | 83.7 (90.6)  |
| RF (n=10, d=20) | 94.7 | 67.5 (71.3) | 71.5 (75.4)  |
| Avg.          | 100.0 | 70.2        | **75.5**     |

Table 5: Breakdown results on CIFAR-10 dataset.

|            | REAL  | WGAN        | BWGAN       |
|------------|-------|-------------|-------------|
| CNN        | 70.8  | 63.5 (89.8) | 63.0 (89.1) |
| Resnet56v2 | 77.5  | 48.8 (62.9) | 51.3 (66.2) |
| Avg.       | 100.0 | 76.4        | **77.6**    |

## 6 DISCUSSION

We introduce an auxiliary loss in GANs which improves the model compatibility of generated dataset. We prove the new loss is unbiased and is applicable to all variants of GAN to improve model compatibility. We further demonstrate that our method has clear advantages with a variety of machine learning models trained on generated dataset. In addition, we investigate the results of feature selection and found that the BC-loss doesn't effect the interpretation of machine learning models. While this work only focus on classification problem, generating data for regression problem is also worth studying. We hope our work will open the path for GANs with better model compatibility so that synthetic data can be more useful in practice.

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

# A DATASET INFORMATION

Table 6: Attributes of UCI datasets

| Dataset | # of train | # of test | # of discrete feature | # of continuous feature | # of class |
|---|---|---|---|---|---|
| Adult | 32561 | 16281 | 123 | 0 | 2 |
| Connect-4 | 54046 | 13511 | 126 | 0 | 3 |
| Covertype | 116203 | 116202 | 44 | 10 | 7 |
| Sensorless | 46807 | 11702 | 0 | 48 | 11 |

# B DETAIL RESULT

## B.1 ADULT

| | REAL | ACGAN | WGAN | BWGAN | MMDGAN | BMMDGAN |
|---|---|---|---|---|---|---|
| DT (d=10) | 83.6 (100.0) | 80.8 (96.6) | 80.3 (96.0) | 81.9 (97.9) | 79.7 (95.3) | 81.2 (97.1) |
| DT (d=20) | 81.2 (100.0) | 80.9 (99.6) | 75.9 (93.5) | 79.9 (98.4) | 73.8 (91.0) | 75.9 (93.5) |
| Linear SVM | 81.2 (100.0) | 80.9 (99.6) | 75.9 (93.5) | 79.9 (98.4) | 73.8 (91.0) | 75.9 (93.5) |
| MLP (100) | 84.4 (100.0) | 81.8 (96.9) | 83.1 (98.5) | 83.6 (99.1) | 82.9 (98.2) | 84.1 (99.7) |
| MLP (200x2) | 84.4 (100.0) | 82.0 (97.2) | 83.1 (98.5) | 83.8 (99.3) | 82.6 (97.9) | 84.2 (99.8) |
| RF (n=10, d=10) | 83.9 (100.0) | 82.6 (98.4) | 83.4 (99.4) | 83.6 (99.6) | 83.5 (99.5) | 83.3 (99.3) |
| RF (n=10, d=20) | 84.1 (100.0) | 80.7 (96.0) | 81.5 (96.9) | 83.2 (99.0) | 81.5 (96.9) | 82.3 (97.8) |
| Avg. | 100.0 | 97.8 | 96.6 | 98.8 | 95.7 | 97.2 |

## B.2 CONNECT4

| | REAL | ACGAN | WGAN | BWGAN | MMDGAN | BMMDGAN |
|---|---|---|---|---|---|---|
| DT (d=10) | 74.7 (100.0) | 64.1 (85.8) | 67.7 (90.7) | 69.5 (93.1) | 66.7 (89.3) | 68.5 (91.7) |
| DT (d=20) | 76.3 (100.0) | 64.1 (83.9) | 60.9 (79.8) | 64.9 (85.1) | 61.4 (80.4) | 64.1 (84.0) |
| Linear SVM | 76.3 (100.0) | 64.1 (83.9) | 60.9 (79.8) | 64.9 (85.1) | 61.4 (80.4) | 64.1 (84.0) |
| MLP (100) | 84.2 (100.0) | 66.0 (78.4) | 74.7 (88.7) | 73.7 (87.6) | 72.2 (85.8) | 72.8 (86.5) |
| MLP (200x2) | 85.7 (100.0) | 66.6 (77.7) | 74.5 (86.9) | 73.7 (86.0) | 71.9 (83.8) | 72.9 (85.0) |
| RF (n=10, d=10) | 73.0 (100.0) | 66.8 (91.6) | 71.4 (97.8) | 70.4 (96.4) | 70.1 (96.0) | 68.3 (93.6) |
| RF (n=10, d=20) | 79.2 (100.0) | 67.0 (84.6) | 70.7 (89.3) | 70.8 (89.4) | 69.8 (88.1) | 68.9 (87.0) |
| Avg. | 100.0 | 83.7 | 87.6 | 88.9 | 86.3 | 87.4 |

## B.3 COVERTYPE

| | REAL | ACGAN | WGAN | BWGAN | MMDGAN | BMMDGAN |
|---|---|---|---|---|---|---|
| DT (d=10) | 77.1 (100.0) | 37.8 (49.1) | 65.6 (85.1) | 68.9 (89.4) | 65.5 (84.9) | 66.7 (86.5) |
| DT (d=20) | 86.9 (100.0) | 39.4 (45.3) | 62.9 (72.3) | 67.2 (77.3) | 62.2 (71.6) | 63.4 (73.0) |
| Linear SVM | 86.9 (100.0) | 39.4 (45.3) | 62.9 (72.3) | 67.2 (77.3) | 62.2 (71.6) | 63.4 (73.0) |
| MLP (100) | 80.6 (100.0) | 54.0 (67.0) | 67.5 (83.8) | 69.9 (86.7) | 62.5 (77.5) | 66.2 (82.2) |
| MLP (200x2) | 89.2 (100.0) | 53.6 (60.1) | 66.5 (74.5) | 68.3 (76.5) | 58.8 (65.9) | 65.1 (73.0) |
| RF (n=10, d=10) | 73.8 (100.0) | 38.6 (52.2) | 67.2 (91.0) | 69.5 (94.1) | 68.0 (92.1) | 67.8 (91.8) |
| RF (n=10, d=20) | 86.2 (100.0) | 38.6 (44.8) | 67.2 (77.9) | 69.7 (80.8) | 65.9 (76.4) | 68.3 (79.2) |
| Avg. | 100.0 | 52.0 | 79.6 | 83.2 | 77.1 | 79.8 |

## B.4 SENSORLESS

|  | REAL | ACGAN | WGAN | BWGAN | MMDGAN | BMMDGAN |
|---|---|---|---|---|---|---|
| DT (d=10) | 96.3 (100.0) | 75.8 (78.7) | 77.9 (80.8) | 87.2 (90.5) | 82.3 (85.5) | 82.6 (85.8) |
| DT (d=20) | 98.4 (100.0) | 75.4 (76.7) | 68.3 (69.5) | 89.6 (91.1) | 81.3 (82.6) | 83.8 (85.2) |
| Linear SVM | 98.4 (100.0) | 75.4 (76.7) | 68.3 (69.5) | 89.6 (91.1) | 81.3 (82.6) | 83.8 (85.2) |
| MLP (100) | 93.6 (100.0) | 72.8 (77.8) | 87.8 (93.8) | 89.8 (96.0) | 82.4 (88.0) | 84.9 (90.6) |
| MLP (200x2) | 98.7 (100.0) | 76.8 (77.8) | 90.5 (91.7) | 93.7 (94.9) | 85.4 (86.5) | 87.7 (88.8) |
| RF (n=10, d=10) | 98.4 (100.0) | 76.2 (77.5) | 92.1 (93.6) | 92.5 (94.1) | 87.9 (89.4) | 88.7 (90.2) |
| RF (n=10, d=20) | 99.8 (100.0) | 77.1 (77.3) | 93.3 (93.6) | 95.6 (95.8) | 89.2 (89.4) | 90.8 (91.0) |
| Avg. | 100.0 | 77.5 | 84.6 | 93.3 | 86.3 | 88.1 |

