# OpenReview forum: "Improving Model Compatibility of Generative Adversarial Networks by Boundary Calibration"
_ICLR.cc/2020/Conference — Reject_

### Official Review · AnonReviewer1 · 2019-10-21
**Official Blind Review #1**

**Rating:** 3

**Review:**

In this work authors consider a problem of 'model compatibility' of GANs, i.e. usefullness of the generated samples for classification tasks. Proposed 'Boundary Calibration' GAN attempts to tackle this issue by adding non-adversarial terms to discriminator, obtained as outputs of the classifiers trained on the original data. For evaluation, it is proposed to compare accuracies obtained by classifiers trained on generated and on real data (termed 'relative acurracy'). Experiments show that the proposed methods improve such scores.

Pros:
- considered problem seems to be important GAN application which has not yet received too much attention.
- proposed method seems to improve the accuracy of classifiers trained on generated data.

Cons:
- the paper is poorly written, has multiple typos and often it is unclear what authors mean.
- the proposed evaluation does not exactly measure the potential improvements from training classifiers with generated data. It would make sense to provide information if generated data can improve classifier's scores, if added to the training data (or small part of it).
- it is unclear whether or not the proposed technique does not affect the sample quality, as no quantitative metrics are provided.

Details:
1. Quantitative scores for quality of generated samples are not provided. Authors instead provide few samples and state that it is difficult to detect the difference. Although sample quality is not the main task here, it certainly is important - otherwise we could train generators solely against the classifier 'boundary calibration' loss terms - this, however would likely lead to adversarial examples. Combining two losses often leads to trade-offs, hence showing that we can improve 'model compatibility' without the loss of sample quality is actually crucial. Metrics such as Inception score [1], FID [2] or KID [3] are desired, especially given relatively poor quality of cifar and mnist samples from all models.
2.The main metric used is the ratio between accurracies of classifiers (trained on real and generated data). It is hence difficult to tell if the classifiers that were used were trained reasonably and achieved reasonable scores.
3. In the abstract, authors claim that 'GANs often prefer generating easier synthetic data that are far from boundaries of the classifiers'. Although for some GAN settings the generators might be biased to do so, in general this claim is unfounded, as GANs optimize divergences that are agnostic to classifier boundaries.
4. It is unclear what kind of classifier-output is used as an input to MMD. Are these continuous logits, discrete class numbers, or one-hot-encoded class identities?
5. Authors use WGAN and MMDGAN with gradient penalty. It is unclear how gradient penalty is applied to MMDGAN as what should be penalized is the witness function, which is different than in WGAN-GP [4], see e.g. [3].
6. It is unclear how embeddings of class information are concatenated to discriminator inputs (p.5).
7. It is unclear to what extent feature selection is deterministic. Authors argue in Section 5.4 that the intersection of top-k features selected from two models should be large. It would be good to provide the same statistics for features selected twice on the same sample.

Overall, the paper currently does not match the quality requirements of ICLR, however it has potential for improvement if the mentioned issues are addressed.

Typos/unclear expressions:
[p1] 'may not willing' >> 'may not be willing'
[p1]'with the property similar to the original data is demanding'  >> properties, demanded/in demand
[p2] 'Although GANs are versatile as aforementioned' - strange wording
[p2] 'The pioneered work' >> 'pioneering work'
[p2] 'information of models' >> 'information from the models'
[p2] effects >> affects
[p3] related works >> related work
[p3] distribution of label >> distribution of labels
[p3] 'generated dataset adopt ' >> 'generated dataset will adopt '
[p3] 'To known about the boundary' >> ' To know the boundary'/'To include the information about the boundary'
[p3] 'the a distance'
[p3] 'the problem to distinguish whether two sets of samples'
[p4] 'If they are close the sets might be sampled from the same distribution' (?)
[p4] tries to minimized the MMD
[p4] would not leads to
[p6, Table 2 caption] number of estimator used
[p6] at Appendix
[p6] depresses the model compatibility (?)
[p6] can providing
[p7] to known how
[p8] our work open >> our work will open


------------------------------------- Revision -------------------------------------

Although some issue seem to have been clarified, two of my main concerns, i.e. the proposed evaluation and the text quality, have not been resolved. Additionally, as pointed out by reviewer #4, the results seem somewhat incremental. For these reasons, I decided to keep the rating unchanged.

**Experience Assessment:**

I have published one or two papers in this area.

**Review Assessment: Checking Correctness Of Derivations And Theory:**

I carefully checked the derivations and theory.

**Review Assessment: Checking Correctness Of Experiments:**

I carefully checked the experiments.

**Review Assessment: Thoroughness In Paper Reading:**

I read the paper thoroughly.

---

> ### Author Response · Authors · 2019-11-15
> **Response to Reviewer #1**
>
> Thank you for the detailed feedback and constructive advice. We’ve revised our paper according to the suggestions. Below we detail each comment individually.
>
> Q1: “Quantitative scores for quality of generated samples are not provided.”
> A1: Thank you for the comment. We’ve added the Inception score and FID for CIFAR-10 in the captions of Figure 2 in our revision.
>
> Q2: “The main metric used is the ratio between accuracies of classifiers (trained on real and generated data). It is hence difficult to tell if the classifiers that were used were trained reasonably and achieved reasonable scores.”
> A2: We use the ratio instead of using the raw accuracy because it would be more reasonable to provide a summary score for different models. All classifiers are trained until the validation loss converged. We have also provided the testing accuracies for each experiment in Appendix for reference.
>
> Q3: “In the abstract, authors claim that 'GANs often prefer generating easier synthetic data that are far from boundaries of the classifiers'. Although for some GAN settings the generators might be biased to do so, in general this claim is unfounded, as GANs optimize divergences that are agnostic to classifier boundaries.”
> A3: Thank you for pointing out this issue. We agree that the claim is too arbitrary. We’ve modified the abstract to better describe the fact.
>
> Q4: “It is unclear what kind of classifier-output is used as an input to MMD. Are these continuous logits, discrete class numbers, or one-hot-encoded class identities?”
> A4: Thank you for the question. In practice we use softmax to obtain the posterior of classifiers. We’ve added some explanation in Section 4.1 to make it clearer.
>
> Q5: “Authors use WGAN and MMDGAN with gradient penalty. It is unclear how gradient penalty is applied to MMDGAN as what should be penalized is the witness function, which is different than in WGAN-GP [4], see e.g. [3].”
> A5: The usage of gradient penalty in MMDGAN has been described in [1] and [2]. We follow the implementation of [2], which is available at the Github repository [3].
>
> Q6: “It is unclear how embeddings of class information are concatenated to discriminator inputs (p.5).”
> A6: Thank you for pointing out the problem. The embeddings of class information are concatenated as additional features to discriminator inputs. We’ve added the explanation in Section 5.1 to describe it explicitly.
>
> Q7: “It is unclear to what extent feature selection is deterministic. Authors argue in Section 5.4 that the intersection of top-k features selected from two models should be large. It would be good to provide the same statistics for features selected twice on the same sample.”
> A7: Thank you for the suggestion. We’ve added a column in Table 3 and Table 4 to show the statistics for feature selected on the same sample with a different random seed.
>
> Typos: We appreciate the reviewer for reading the paper thoroughly and pointing out the typos. We’ve corrected them in the revision and checked the paper again.
>
>
> [1] Chun-Liang Li, Wei-Cheng Chang, Yu Cheng, Yiming Yang, Barnabás Póczos: MMD GAN: Towards Deeper Understanding of Moment Matching Network. NIPS 2017: 2203-2213
> [2] Mikolaj Binkowski, Dougal J. Sutherland, Michael Arbel, Arthur Gretton: Demystifying MMD GANs. ICLR (Poster) 2018
> [3] https://github.com/MichaelArbel/Scaled-MMD-GAN

---

### Official Review · AnonReviewer2 · 2019-10-23
**Official Blind Review #2**

**Rating:** 3

**Review:**

This paper aims at training a GAN that can generate data matches the real data distribution well especially at the boundaries of the classifiers. A Boundary-Calibration loss (BC-loss) base on multi pretrained classifiers is introduced to match the statistics between the distributions of original data and generated data. The motivation is interesting. The story is clearly explained. However, the experiments part is weak.

There are several typo and mistakes. The experiments only show that the proposed method got a good performance, but the analysis of the reason is not shown. The reason to name the loss as Boundary-Calibration loss (BC-loss) should be explained and the experiments should show some effect on the boundary areas. Some concerns are listed below,

1.	If the generated dataset exhibits very good property as the real dataset, it means the data is to some extent perfectly foreseen, and there is little to no privacy, is it contrary to the aim of not leaking the real dataset?
2.	It is more interesting to see the difference between the distribution of real data and the generated data, However the author only show a simple toy data distribution comparison, I would like to see more comprehensive results about the distribution differences on real dataset , e.g. the TSNE embedding?
3.	Equation (3) is not correct.
4.	The author said that image quality of MNIST and CIFAR10 are not improved, then why the classification results are improved?  there should have some differences existed among different compared methods, it would be more convincing if you can show it out.
5.	What kind of  generator do you use for the UCI data?  How do you settle the output problem?  Since some of the data are continuous and some are discrete.


**Experience Assessment:**

I have published one or two papers in this area.

**Review Assessment: Checking Correctness Of Derivations And Theory:**

I assessed the sensibility of the derivations and theory.

**Review Assessment: Checking Correctness Of Experiments:**

I assessed the sensibility of the experiments.

**Review Assessment: Thoroughness In Paper Reading:**

I read the paper at least twice and used my best judgement in assessing the paper.

---

> ### Author Response · Authors · 2019-11-15
> **Response to Reviewer #2**
>
> Thank you for the valuable comments. We’ve revised our paper according to the suggestions. Below we detail each comment individually.
>
> Q1: “If the generated dataset exhibits very good property as the real dataset, it means the data is to some extent perfectly foreseen, and there is little to no privacy, is it contrary to the aim of not leaking the real dataset?”
> A1: Thank you for the valuable comment. We agree that a perfect generator may leak the real dataset to some extent. However, it depends on the support size of the real dataset, which is hard to evaluate. For example, it is hard to not generate the same instance for a low-dimension discrete dataset. Besides, previous work[1] has shown that empirically, GANs are lack of diversity (low support size) instead of memorizing training set.
>
> Q2: It is more interesting to see the difference between the distribution of real data and the generated data, However the author only show a simple toy data distribution comparison, I would like to see more comprehensive results about the distribution differences on real dataset , e.g. the TSNE embedding?
> A2: Thank you for the suggestion. We agree that showing the distribution differences on a real dataset will make this work more comprehensive. We’ve tried to visualize the distributions by PCA and T-SNE. However, for UCI datasets, we can't even observe clear clusterings for different classes of real data, so we used a well-trained fully-connected network with a 2-units hidden layer before the output layer to project the generated samples to a 2-dimensional embedding space. We've added the result and discussion in Section 5.3.
>
> Q3: Equation (3) is not correct.
> A3: Thank you for pointing out the typo. We’ve corrected it in the revision.
>
> Q4: The author said that image quality of MNIST and CIFAR10 are not improved, then why the classification results are improved? there should have some differences existed among different compared methods, it would be more convincing if you can show it out.
> A4: Thank you for the comment. We agree that there should be some reasons except image quality that makes the classification accuracy improved. We assume that generating more images with correct labels can help a classifier learn better. Unfortunately, we do not find a good way to show it clearly so far.
>
> Q5: What kind of generator do you use for the UCI data? How do you settle the output problem? Since some of the data are continuous and some are discrete.
> A5: Thank you for the question. In our experiment, discrete features are processed to one-hot encoding and continuous features are scaled to [0, 1]. Therefore the features can be generated by a logistic function.
>
>
> [1] Sanjeev Arora, Andrej Risteski, Yi Zhang: Do GANs learn the distribution? Some Theory and Empirics. ICLR (Poster) 2018

---

### Official Review · AnonReviewer4 · 2019-11-19
**Official Blind Review #4**

**Rating:** 3

**Review:**

In this paper the authors propose a method for improving "model compatibility" in GANs. For this reason they add to the loss of the generation procedure a term that depends on the maximum mean discrepancy between the following datasets: (1) the output of a classifier with input the real dataset, (2) the output of the same classifier with input GAN-generated samples. They authors show that in essentially all the datasets they tried, the model compatibility of the produces generator is increased after adding the aforementioned cost, while the visual quality of the data is not decreased.

Strengths:

- The low model compatibility of GANs is a very important disadvantage and hence improving this aspect of GANs is a relevant problem.

Weaknesses - Comments:
A. The increase in the model compatibility is very mild. Especially in CIFAR-10, the increase in very small.

B. In MNIST the increase is larger than CIFAR-10 but the initial model compatibility using vanilla GANs is smaller. The reason might be that for MNIST much simpler classification algorithms have been used. This maybe suggests that the proposed method affects more the model compatibility of methods that achieve lower model compatibility in before the addition of the extra cost term.

Minor Comments:
1. In equation (1) it looks strange that the summation is over A but A does not appear at all in the summand. I suggest you replace h and h' with A(D) and A(D') so that this is clear.
2. In Theorem 2, \hat{L}_G is used but for the proof the authors have replaces \hat{M} with M. There should be a comment for that. In general I believe that Theorem 2 is almost trivial and does not add value to this clearly experimental paper.

**Experience Assessment:**

I have read many papers in this area.

**Review Assessment: Checking Correctness Of Derivations And Theory:**

I assessed the sensibility of the derivations and theory.

**Review Assessment: Checking Correctness Of Experiments:**

I did not assess the experiments.

**Review Assessment: Thoroughness In Paper Reading:**

I made a quick assessment of this paper.

---

### Decision · Program_Chairs · 2019-12-19

**Decision:**

Reject

**Comment:**

The paper presents a method for increasing the "model compatibility" of Generative Adversarial Networks by adding a term to the loss function relating to classification boundaries. The reviewers recognized the importance of the problem, but several concerns were raised about the clarity of the paper, as well as the significance of the experimental results.